# Clinicians' views on treatment adaptations for men with eating disorders: a qualitative study

Emma Kinnaird,[1] Caroline Norton,[2] Kate Tchanturia[1,2,3]

EK and CN are joint first authors

[1]Department of Psychological Medicine, Institute of Psychiatry, Psychology and Neuroscience, King's College London, London, UK
[2]Eating Disorders National Service, South London and Maudsley NHS Foundation Trust, London, UK
[3]Department of Arts and Sciences, Ilia State University, Tbilisi, Georgia

**Correspondence to**
Dr Kate Tchanturia;
kate.tchanturia@kcl.ac.uk

## ABSTRACT

**Objectives** Despite traditional views of eating disorders as a female illness, there is a growing body of evidence that the incidence rate of eating disorders in men is rising. Research suggests that these men may experience unique symptoms and difficulties, however, it is unclear how these unique needs may impact treatment. The aim of this study was to explore clinicians' views on whether men have gender-specific treatment needs, and how far these needs require treatment adaptations.

**Design** Qualitative interview study using framework analysis to explore the experiences of clinicians working with men with eating disorders.

**Setting** Outpatient National Health Service eating disorder service in London.

**Participants** Ten clinicians from a variety of clinical backgrounds participated in the study.

**Results** The following three themes emerged: male-specific issues identified by clinicians, treatment approaches used for this population and the importance of creating a male-friendly environment. Male-specific issues identified by participants included an increased focus on muscularity and difficulty expressing or discussing emotion. Clinicians also suggested that men may be more likely to adopt a performance-based approach to. This was linked by clinicians to the impact of cultural perceptions of masculinity on their patients. Clinicians in this study felt that these individual needs could be met by adapting existing approaches within a supportive, male-friendly environment. However, there was not consensus over specific adaptations, including identifying risk, the need for male-only groups, or whether male patients needed access to male clinicians.

**Conclusions** Although men do present with specific treatment needs, these can typically be met within the framework of typical treatment approaches by experienced clinicians in an environment sensitive to the presence of men in an otherwise female-dominated space. However, there are a lack of explicit guidelines for this process, and areas such as male-only treatment spaces require further research.

## INTRODUCTION

Eating disorders (EDs) have traditionally been perceived as an illness affecting young women. However, there has been a significant increase in research interest in men with EDs since studies began to suggest that men,

### Strengths and limitations of this study

► Exploration of clinician views on an under-researched and increasingly relevant area. Only included clinicians from a London National Health Service service so may not be generalisable to other jurisdictions in the UK and beyond.
► Only interviewed clinicians working in a mixed gender eating disorder service, so clinicians may not have had experience of working in a male-only treatment centre.
► Raises concepts that can be explored using further empirical research.

far from being a diagnostic rarity, potentially accounted for up to 10% of individuals with EDs.[1] More recent research suggests that men could represent as many as one in five people with EDs in the UK, with this number rising to one in four people with EDs in the USA.[2 3] With incidence rates of EDs in men rising, effectively treating men with EDs is becoming a growing priority.[4]

There has been an increased recognition of unique symptoms and issues experienced by men with EDs that may translate into unique treatment needs.[5] Men with EDs typically have a later age of onset than women, are more likely to report a previous history of being overweight or obese and are more likely to present with psychiatric comorbidities, including substance abuse, psychosis or personality disorder.[6 7] Moreover, men with EDs appear to experience different symptoms to women likely moderated by sociocultural factors surrounding masculinity and body image.[8] These include less of a desire to lose weight, a greater focus on exercise rather than vomiting or laxatives as a form of purging and specific sexual difficulties.[9–12] Differences in symptoms appear to be influenced by a greater drive towards muscularity in men, compared with a drive for thinness.[13–15] In addition, men with EDs may pose additional physical risks, with the

severity of physical symptoms at the time of treatment presentation potentially heightened due to delays in accessing treatment.[16] Research suggests that using body weight as an indicator of severity and medical risk may be less reliable in male patients, with men potentially exhibiting a higher body weight than their female counterparts, yet also being at greater risk for osteoporosis and bone disease.[17 18] However, at present, no specific clinical guidelines exist for the physical assessment of men with EDs.[5]

The issue of whether men require gender-specific treatment adaptations remains controversial: men with EDs themselves indicate that they may experience different issues to women in treatment,[19] and previous research has suggested gender-specific treatment adaptations to aid recovery in male patients, including male-only groups.[11] In particular, a survey carried out on service provision for men with EDs in Australia raised a number of adaptations used by practitioners, including the importance of challenging masculinity, the possibility of male-only groups and differences in emotion processing.[20] By comparison, other studies have reported that male ED recovery experiences and requirements are highly similar to those of women, with men in fact more likely to achieve better treatment outcomes than women, indicating that treatment adaptations may not be necessary.[21–23] Therefore, there is a significant lack of consensus in this area, with previous qualitative research examining the views of men themselves similarly finding disagreement on whether their gender should be considered as a relevant factor in treatment.[24]

Effectively treating male EDs is becoming a growing priority in the National Health Service (NHS), with the Joint Commissioning Panel for Mental Health recommending that 'gender appropriate services should be available to all.'[25 26] However, the question of what exactly gender appropriate services represent is still uncertain: when men do access treatment in the UK, it is unclear how far current approaches used by clinicians require adaptation for this population. While previous literature has reflected on the need male treatment adaptations, these have not been empirically explored and rather represent clinical recommendations.[27] Consequently, this paper aimed to explore this area by focusing on the following research questions:

1. Do clinicians believe that men with EDs have gender-specific issues?
2. Do clinicians believe that men with EDs require gender-specific treatment adaptations?

## METHODS
### Design
This study used a qualitative design, employing semistructured interviews exploring the potential of developing treatment adaptations for male EDs. The study is reported according to Consolidated Criteria for Reporting Qualitative Research guidelines.[28]

### Setting
Clinicians were recruited from the South London and Maudsley NHS Foundation Trust National Eating Disorders Service. At the time of interviews (January to February 2017), the service had a total of 491 patients, of which 58 (12%) were male.

### Participants
The sample consisted of clinicians currently working within the outpatient and day-care teams treating adults with EDs, with a minimum 3 years of experience in this area. Clinicians all had experience of treating male EDs. Clinicians were invited to take part in an interview assessing their views on treating male EDs through email. All individuals agreed to participate, representing 55% of the clinicians in the department and so formed the final cohort. Participants came from a range of clinical backgrounds. All participants were female, reflecting the all-female staff demography of clinicians within the department. Participants were informed that the interviews were part of a project aimed at improving service provision for men with EDs.

### Data collection
All interviews took place at the clinician's place of work. Interviews were carried out by CN, the female team leader of the outpatient unit with a background in nursing and EDs. While the impact of participants being interviewed by a senior staff member was a concern for the research team, steps were taken to minimise this as an issue: interviews were held in a room in the hospital separate from the department, and CN discussed the study with participants prior to the actual interview. Moreover, it was emphasised that this was a service improvement project, with an emphasis on service provision rather than assessing the participants as individuals. Written consent was acquired prior to interviews, including consent to audio record. Participants were asked the following questions:

1. Do you work differently with men and women with EDs?
2. When treating men, do you look for any male-specific issues during treatment?
3. What do you think our strengths are in our service treating men with EDs?
4. What do you think could be improved?

The interviewer then asked follow up questions based on themes that arose during the interviews, and anticipated themes based on previous research literature. Interviews lasted between 20 and 40 min, and were audio recorded. Field notes were additionally made during the interview by EK. Recordings were then transcribed with any identifying information removed at the point of transcription. Transcripts were not returned to participants. Following the interviews with the 10 participants who had first agreed to participate, it was judged by the authors that data saturation had been reached as no new information was seen to be emerging from the interviews.

## Analysis

Data were analysed using framework analysis.[29] Interview data were entered into NVivo V.11 for data management and coding. A coding framework was developed deductively based on the research aims, questions asked during the interviews and previous literature. The coding framework was then further inductively refined based on interview data content. This gave rise to a coding framework consisting of four main categories focusing on male/female symptom differences, male/female treatment differences, need for male treatment adaptations and service improvements. All authors met to achieve consensus on these categories, and this framework was then applied to the data by EK. Coded data were then analysed to identify themes relevant to the research question. Three themes emerged from the analysis following coding: male-specific issues, treatment approaches and creating a male-friendly environment. Themes are reported together with supporting quotes, anonymised using participant numbers.

## Patient involvement

The development and design of this study was informed by a patient steering group consisting of men who had received treatment for EDs in the service. The notion that men may require treatment adaptations was initially explored with this group, with their responses leading to the development of the service improvement project. The interview schedule for this study was based on issues raised and explored by this male patient group.

## RESULTS

### Male-specific issues

Participants described how although symptoms were broadly similar across genders, men sometimes presented with distinct features, including an increased focus on muscularity. There appeared to be a link between the perceived emphasis of these male patients on exercise and fitness, and a tendency to perceive their illness in more 'mechanical' or functional terms: one participant described how her male patients were more likely to have first made contact with health services due to physical injuries, which led to the diagnosis of an ED, rather than seeking help for emotional distress (participant 2). Where male patients did experience negative emotions, this tended to be perceived as a failure of masculine ideals, rather than in emotional terms.

> Women feel shame and embarrassment with binges and over-eating and all that, men feel less so when they are in company because its more macho to eat more. I don't think they have a problem with that until it feels out of control. (Participant 1)

This was consistent with a common observation across participants that they found it more difficult to encourage their male patients to talk about their emotions compared with female patients. This was again related to the pressures of masculine cultural ideals.

> Only certain emotions are encouraged in men, like it's ok to be angry, it's ok to be tough, you need to be stoic and you're not allowed to be, you know, weak and vulnerable… part of being a man a lot of the time is being able to deal with things and be stoic and just man up and I think actually talking about things isn't always encouraged. (Participant 2)

This raised the concept that for male patients, engaging in ED treatments that require discussing emotions, particularly talking therapy, could itself be seen as a challenge to their masculinity.

A number of clinicians clarified, however, that their observations on male-specific issues were not generalisable to all men, and that symptoms such as an increased focus on muscularity were not necessarily limited to one gender. In particular, the emotional dysregulation symptoms associated with binge eating disorder (BED) were perceived to be similar across both men and women, enabling clinicians to approach treating individuals in a similar way.

> I think I'm doing a lot of the same work with binge eaters, yeah—I think a lot of it is focused on getting them to connect with how they feel. I think that once they've done that there's a major turning point for both females and males. So I don't think that there's anything gender specific that I'm doing with them. (Participant 2)

Consistent with this point, clinicians additionally highlighted that difficulty expressing emotion was not limited to male patients, and in fact was common across patients with anorexia nervosa (AN).

### Treatment approaches

Although participants described a number of male-specific issues, the majority of clinicians suggested that they would not fundamentally approach treating men any differently to women. Instead, gender was one of a number of individual factors they considered when approaching treatment.

> I wouldn't necessarily do something different just because the person sitting in front of me was a man or a woman, it's really what they bring and then just using the same models that I use for everybody tailored to the individual rather than the gender or the sexuality or the race. (Participant 4)

From this perspective, they emphasised that a key aspect of therapeutic treatment was that it could be adapted to meet these issues and problems raised by individuals, including gender-specific elements, and suggested that this approach represented a strength of the ED service in treating men.

Consequently, there was a consensus that male needs could be accommodated within the individualised,

flexible nature of normal treatment approaches. Specific adaptations within this treatment framework described by clinicians to meet these specific needs included additional work on education surrounding ED stigma, and a greater focus on emotional expression and identification. In the context of an observed male reluctance to express emotion, or show vulnerability, clinicians suggested that they would proactively raise certain difficulties commonly experienced by men with their male patients, rather than relying on their patients to raise these issues themselves: 'I think at the moment it's about us having to keep asking him questions rather than him being able to come to us' (participant 10).

Therefore, clinicians often perceived a key element of these treatment adaptations as challenging traditional ideas of masculinity. Clinicians felt that men, consistent with motivations behind their symptoms, approached recovery from a mechanical or performance-based perspective which resulted in them attempting to 'eat their way out' of recovery without addressing the underlying emotions, or challenging damaging masculine ideals.

> We could just let him eat his way out, but that's recovery he's done twice before and he's relapsed on both occasions and that's exactly the difference that we were talking about- could this time be different because maybe there's something about, maybe supporting him just to know how to actually ask for help… Because his mum said that on the previous two occasions when he has relapsed it's happened really quickly and he hasn't flagged up that he's having any difficulties. (Participant 10)

Consequently, actively challenging masculine ideals surrounding emotion, vulnerability and performance, was perceived as fundamental to ensuring the success of traditional, emotion focused, treatment approaches for male patients.

The majority of clinicians in this study had extensive experience working with male patients, and felt comfortable and confident in discussing how they adapted treatment for these men, and how they would proactively raise issues that they knew to be particularly relevant to treatment for men with EDs. However, this flexible, informed approach appeared to stem from their previous experience working with men, rather than any previous training. By contrast, a minority of clinicians with less experience working with male patients suggested that they would feel less confident making these kind of treatment adaptations, and indicated a desire for greater training in this area. Additionally, although clinicians were able to discuss the kinds of therapy adaptations they would make when working with men, the majority of participants suggested that they felt less comfortable in managing the physical aspects of male EDs.

> I think that the physical stuff is quite difficult with men. I know less about the physical impacts on men and things like BMI ranges and what I'm looking out for physically with men. I feel more confident knowing the physical, medical side of things with women. (Participant 6)

This to an extent reflected variance in clinical backgrounds across participants, with participants with training in nursing or nutrition exhibiting greater confidence in these clinical aspects of treatment than those with backgrounds in therapy. However, there was disagreement across participants surrounding whether or not male EDs were associated with greater risks. A minority of clinicians suggested that they would be concerned that their male patients were at greater risk from behaviours such as self-harm or suicide, but were unsure how to address this issue in treatment.

### Creating a male-friendly environment

Clinicians described the perceived significance of cultural masculine ideals to their male patients, and the impact of these ideals both on treatment needs compared with women, and the way in which men engaged with treatment. From this perspective, clinicians emphasised that rather than fundamentally altering treatment approaches for men with EDs, it was instead important to deliver this treatment within an environment sensitive to the presence of men within a female-dominated service and a female-dominated illness.

Clinicians suggested that EDs and ED treatment were perceived by men (and wider society) as inherently anti-masculine: 'they feel as though it's a female disease' (participant 1). Therefore, they felt that a key element of effectively treating male EDs was the importance of challenging, and not subconsciously reinforcing these perceptions through the process of treatment. This involved raising the issue of wider societal stigma surrounding EDs with their patients, but also acknowledging that this perception of EDs as a female illness was potentially reinforced by the nature of the treatment service: participants were conscious that the service had an all-female staff and a majority of female patients.

> I think it must be very hard for men walking through the door because they are going to be sat in a waiting room full of women, and that's going to be their perception of coming, and then it's played out when they arrive. And also we are a department which doesn't seem to have any male therapists and I am sure all of those things are difficulties. (Participant 3)

Therefore, clinicians described that while normalising male EDs was a key part of treatment, they tried to additionally reinforce this process by adapting the surrounding environment. This included putting up posters in the waiting room about male EDs, and altering therapeutic materials to include both male and female images and body issues.

Within this context of creating an environment supportive of men with EDs, a number of clinicians felt

it was important to create spaces to discuss male-specific issues by recruiting more male clinicians, and introducing male-only treatment groups. However, this was controversial, with participants disagreeing on whether men felt more comfortable discussing male-specific issues with other men. One common reason given by clinicians for creating male-only groups was the perception that their male patients felt awkward because 'women are so much more emotional' (participant 1). However, the perceived benefit of isolating men from this environment contrasts with the beliefs of other clinicians that exposing men to such an environment emphasising emotional expression was key to treatment and recovery.

## DISCUSSION

The findings of this study suggest that, from the perspective of the clinicians interviewed, men with EDs do have gender-specific treatment needs. However, these features can potentially be met within the framework of typical treatment approaches by clinicians with an awareness of these needs, in the context of a treatment environment sensitive and supportive to men with EDs.

That clinicians in this study felt that men with EDs were likely to present with a number of specific issues, including an increased focus on muscularity and exercise, is consistent with a large body of previous literature in this area.[8 19] Moreover, the finding of this study that clinicians perceived that these needs can be met by standard treatment approaches reflects previous research suggesting that the symptom differences experienced by men with EDs can nonetheless be accommodated within the same models of ED psychopathology as applied to women—the same models targeted by ED treatments.[30 31] This is consistent with previous research on the views of men with EDs suggesting that men and women experience similar challenges in treatment.[19] Therefore, instead of fundamental changes to ED treatment for males, this study emphasised the importance of clinician openness and empathy in adapting the therapeutic approach to these male-specific issues. This resonates with previous research on the views of men with EDs on treatment, which found that, similarly to women, men highlighted the importance of having an empathetic, non-judgemental therapist to having a positive treatment experience.[32]

This general emphasis on clinician openness and empathy reflects wider therapeutic principles that clinicians should be flexible in adapting therapy to individual needs, even while following manualised treatment programmes.[33] However, the clinicians interviewed in this study highlighted a number of adaptations specifically for men with EDs. Consistent with previous research on treatment adaptations for men with mental health problems, specific recommendations involved including more content on emotional expression and identification.[34] That clinicians particularly highlighted the area of emotional expression as a key gender difference between their male and female patients is in line with a large body

of research suggesting that women are typically better at emotion recognition than men, although with significant individual variation across both groups.[35 36]

The concept of challenging ideas of masculinity has previously been identified by practitioners as a particularly important factor in treating male EDs.[20] However, this present study extended this concept by emphasising that challenging ideas of masculinity involved holistic aspects beyond addressing these issues in therapy, highlighting the importance of the treatment environment itself. Previous research has suggested that men with EDs perceive their illness as inherently feminine, and that practitioners should ensure that treatment is inclusive and relevant to men.[19] The present study went further by suggesting that accepting treatment for EDs, as well as the ED itself, may also be perceived by male patients as inherently antimasculine, creating barriers for men engaging in treatment. This resonates with the accounts of men with EDs who documented feeling ostracised in their treatment spaces by staff members and other patients due to being a man.[32] Therefore, this study emphasises the need to create a male-friendly treatment environment to combat this perception. For example, clinicians in this study highlighted the issue of making environmental aspects such as the waiting room more gender neutral. Moreover, with quantitative research documenting that men with EDs may be more concerned with muscularity than an emphasis on thinness, treatment materials for EDs may require adaptation to include these diverse body image issues.[14 15] Moreover, these materials could be adapted to ensure that they are more gender neutral, such as by including examples and images of men.

However, consistent with previous research, the concept of whether creating a safe and sensitive treatment environment for men involved providing access to male clinicians or male-only groups was controversial.[20] Understanding the perceived need for male-only treatment spaces is significant in the context of Department of Health and Mental Health Act Code of Practice guidance stating that NHS organisations should work towards eliminating mixed-sex accommodation on inpatient units, and that women-only day rooms should be provided on mental health units.[37 38] The current study indicates that gender-segregated treatment spaces, rather than helping patients, may in fact be harmful to the therapeutic process: clinicians felt that mixed-gender spaces, whether this meant a female therapist or mixed-gender groups, encouraged men to express emotion, and were concerned that men would feel inhibited in the company of only other men. Significantly, a male-only treatment space may conflict with the importance of challenging masculine norms also identified in this study, and may inadvertently reinforce male perceptions of being ostracised from ED treatment spaces due to not being female.[32] Future research would benefit from specifically exploring these areas in more detail with male patients and their carers.

A key issue raised in this study was that of the relationship between clinician experience and confidence in treating

male EDs. Previous research has raised the importance of clinicians asking 'the right questions' to 'uncover' information or difficulties in men otherwise highly motivated by cultural factors not to disclose these issues,[20] and documenting that male patients may find it difficult to engage with treatment if they perceive that their needs were not understood by the practitioner.[19 32] However, this study highlighted that clinicians who do not have prior experience may be less confident in adapting treatment for men in this way. A specific area of concern was that of physical risk: the majority of clinicians in this study suggested that they were unclear on whether men with EDs had differing physical risk factors to women, reflecting the lack of clinical guidelines in this area.[5] Therefore, this study suggests that clinicians treating men with EDs should potentially be provided with training or guidelines for male-specific issues as part of a systematic approach to treatment.

### Limitations

This study took place in a specific ED service based in London. Therefore, the findings of this study cannot be generalised to ED treatment across the UK. However, the issues raised by the clinicians in this study provide a useful basis for further empirical investigation. In addition, this study interviewed clinicians on their views on male EDs in general, rather than specific ED types: while the clinicians in this study did briefly discuss different ED types, including AN and BED, further research is needed to explore gender differences across EDs. Moreover, the disagreement in this study surrounding whether men prefer to talk to other men highlights another drawback of the sampling approach used: the sample consisted only of female clinicians, speculating about male patient perspectives. This emphasises the need for future research addressing the ideas raised in this study surrounding best clinical practice for male ED patients with male patients themselves. Further empirical research is needed to explore the best treatment for men with EDs, and to evaluate the need for treatment adaptations, in order to create a systematic, empirically validated treatment approach.

### Clinical and research recommendations

The findings of this study suggest that there is a need for a systematic approach to treating male EDs, consisting of clear, empirically supported clinical guidelines, and training and support for clinicians who lack experience with this population. Moreover, considerations of best practice for male EDs should include a consideration of the treatment environment. Future research should further explore the concepts of best clinical practice and treatment adaptations, with male patients themselves, with a specific focus on the issues of creating a male-friendly environment, and the need for gender-specific or gender-segregated treatment.

**Acknowledgements** The authors would like to thank the staff of the SLAM outpatient eating disorders service for their support with this paper, in particular Caroline Lewis for transcribing the interviews. The authors would also like to thank the male patients in the ED service who have contributed to the design of this study.

**Contributors** EK analysed and interpreted the data, and was a major contributor in writing the manuscript. CN collected the data and aided with study design. KT designed and supervised the study and contributed in write-up. All authors read and approved the final manuscript.

**Funding** This paper represents independent research, for which CN and KT received funding from the National Institute for Health Research (NIHR) Biomedical Research Centre at South London and Maudsley NHS Foundation Trust and King's College London. KT would also like to acknowledge financial support from MRC and MRF Child and young adult Mental health—the underpinning aetiology of self-harm and eating disorders. EK received PhD funding for this project through the Medical Research Council Doctoral Training Partnership (MRC DTP) scheme (MR/N013700/1).

**Disclaimer** The views expressed are those of the author(s) and not necessarily those of the NHS, the NIHR or the Department of Health.

**Competing interests** None declared.

**Patient consent** Not required.

**Ethics approval** The study was approved by South London and Maudsley NHS trust governance committee as part of a service improvement project assessing current treatment provision for men with EDs

**Provenance and peer review** Not commissioned; externally peer reviewed.

**Data sharing statement** The datasets used during the current study are available from the corresponding author on reasonable request.

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
