## [Reviewer comments · BMJ Open]

ARTICLE DETAILS

TITLE (PROVISIONAL)	Clinicians' views on treatment adaptations for men with eating disorders: a qualitative study
AUTHORS	Kinnaird, Emma; Norton, Caroline; Tchanturia, Kate

VERSION 1 – REVIEW

REVIEWER	Phillipa Hay Western Sydney University, Australia
REVIEW RETURNED	11-Mar-2018

GENERAL COMMENTS	This is a well conducted qualitative study in an under-researched but important area- particularly with the growth in prevalence of eating disorders in men. The study points to the need for greater training and awareness of issues for men in treatment. My comments are minor. Abstract  1. Suggest the authors report the findings a little more cautiously e.g. using terminology such as : 'the following themes emerged or were identified....' 2. I think in qualitative research participants do not necessarily 'agree', I'd suggest use terms such as "there was not consensus" or similar. Strengths and limitations  1. Suggest "May not be generalizable to other jurisdictions in the UK and beyond 2. Perhaps add as a strength the good response rate to interview 3. Were any clinicians working or experienced in a male only treatment centre in the past? Maybe comment on this Introduction –  1. Perhaps cite epidemiology beyond the UK? E.g. studies by Hudson in the US 2. I believe there may have been a previous qualitative study of clinician views -?de Beer 2012 which could be cited (de Beer, Z., & Wren, B. (2012). Eating disorders in males. In J. Fox & K. Goss (Eds.), Wiley series in clinical psychology. Eating and its disorders (pp. 427-441). : Wiley-Blackwell.http://dx.doi.org/10.1002/9781118328910.ch27) Method  1. Were there any constraints for the team leader who did the interviews – perhaps there could be a reflection on this and if the seniority imposed any constraints? 2. Explain the role of The South London and Maudsley NHS trust governance committee. Was this study approved as part of a quality assurance project? Results  1. A minor point - was thematic saturation reached?
--

	Discussion 1. The statement “the finding of this study that these needs can be met by standard treatment approaches” is inaccurate as this was not a treatment study, rather “the finding of this study was that clinicians perceived that these needs can be met by standard treatment approaches”
--	---

REVIEWER	Prof. Ulrich Voderholzer, MD Medical director, Schoen Clinic Roseneck, Prien am Chiemsee, Germany
REVIEW RETURNED	21-Mar-2018

GENERAL COMMENTS	Review BMJ Open: Clinicians’ views on treatment adaptations for men with eating disorders: a qualitative study Major comments 1. Introduction: In general, the introduction is well written, structured, and provides a reasonable framework for the present study. It is not fully clear, yet, how a qualitative study on clinician opinions could contribute to answer the questions that larger empirical studies left open. As a reader, the break from the last paragraph in the introduction to your aims is rather harsh. Wouldn’t it be more intuitive to interview male ED patients if the aim is to explore specific treatment needs, as done by Robinson et al., 2003. Only a suggestion, but may be you could frame your aims in reference to this study. 2. Method: How were the 10 clinicians chosen? Randomly? Were there any male clinicians in the department (though, we later learn from participant 3 that there are none)? Please specify how the final selection came about. 3. Method, p 7: On what ground was it judged by the authors that data saturation had been reached? 4. Results, general: I expected to find the four themes identified in the analysis as headers in the results section. As I understand, themes 2 and 3 pertaining to treatment differences and possible adaptations were combined in “treatment approaches”. While I think there is nothing wrong with this approach, it somewhat supersedes the structure proposed in the analysis section. I stumbled upon this inconsistency while reading. 5. Discussion, p 12, ln 15: That first sentence in the discussion is an overstatement of your findings. What you could actually conclude based on your data is that (10 selected) clinicians suggest that men have specific treatment needs. Whether men actually have specific needs, can at best indirectly be examined with your study design. Please consider rephrasing. The following paragraph citing several quantitative studies, however, very well highlights how your findings add a new perspective to what is already known. 6. Discussion, p 13, lns 1-7: In this paragraph, I miss the notion that clinician openness and empathy are basic therapeutic skills that are instrumental in any psychotherapy for anybody with any disorder. Gender is only one of a range of characteristics that affect therapy. So would marital status, education, parenthood, language proficiency, cultural background etc. and all afford adaptation of the theoretical framework to the individual. Please elaborate on this aspect and be more specific what adaptations clinicians will need to make for MEN with EDs. 7. Discussion, p 14: The issue of male-only treatment setting is discussed in a balanced manner. It is not fully clear to me, however, how your study could show that segregated treatment settings would
--

	be harmful. I actually agree with this notion and, besides that, it is just not possible to create male-only settings on a larger scale as most therapists are women. Also, I think that men could benefit from having a female therapist, whom they might expect to be more emotional. Hence, I think it would be more worthwhile to discuss here how existing facilities and treatment contents could be adapted to men. You alluded to the issue of body shape. Could you elaborate a bit more on how your study contributes to this aspect? Similarly to the very important paragraph on physical complications. Minor comments  1. Abstract, p 2, ln 5: the sentence should read “incidence rate (...) in men is rising”. 2. Abstract, p 2, ln 18: the sentence is rather long and could be partitioned, as it conveys the main message of the paper. 3. Introduction, p 3, lns 20-21: Style; repetition of “in fact”, which might not be necessary in any of the two sentences. 4. Method, data collection, pp 6-7: the information that the interviews lasted 20 to 40 minutes is given twice 5. Method, analysis, p 7, ln 15: “coded” instead of “coding” data? 6. Method, patient involvement: I think it is a great feature of your study that patients had a saying in it. Could you may be give one or two examples of how these patients informed your study? Also, a paragraph should contain at least two sentences. 7. Results, p 8, ln 13: there are two commas missing, I think: “...not allowed to be, you know, weak and vulnerable...” 8. Discussion: not really a limitation but I think you should mention somewhere that you investigated EDs in general and not separately according to diagnosis. For example, you would expect women with AN to have similar difficulties expressing emotions that men with AN. May be you could speculate in one or two sentences about whether your results might apply more or less to the different EDs.
--	---

VERSION 1 – AUTHOR RESPONSE

Reviewer(s)' Comments to Author:

Reviewer: 1

Reviewer Name: Phillipa Hay

Please leave your comments for the authors below

This is a well conducted qualitative study in an under-researched but important area- particularly with the growth in prevalence of eating disorders in men. The study points to the need for greater training and awareness of issues for men in treatment. My comments are minor.

Many thanks for the positive feedback (no action required).

Abstract

1. Suggest the authors report the findings a little more cautiously e.g. using terminology such as : ‘the following themes emerged or were identified....’

Themes are now summarised in the opening sentence of the “Results” section of the abstract, framed more cautiously as “The following three themes emerged: male specific issues identified by clinicians, treatment approaches used for this population, and the importance of creating a male friendly

environment.” That these are the views of the participants in this study has also been emphasised- e.g. “Male specific issues including identified by participants included an increased focus on muscularity”, “Clinicians in this study felt that these individual needs could be met by adapting existing approaches”

2. I think in qualitative research participants do not necessarily ‘agree’, I’d suggest use terms such as “there was not consensus” or similar.

“Disagreement” has been altered to “there was not consensus”

Strengths and limitations

1. Suggest “May not be generalizable to other jurisdictions in the UK and beyond

Altered using wording recommended by reviewer

2. Perhaps add as a strength the good response rate to interview

Response rate was miscommunicated in initial draft- has been altered to 70%

3. Were any clinicians working or experienced in a male only treatment centre in the past?

Maybe comment on this

Unfortunately we do not have this information on the clinicians’ prior experience. However, have added “Only interviewed clinicians working in a mixed gender eating disorder service, so clinicians may not have had experience of working in a male only treatment centre”

Introduction –

1. Perhaps cite epidemiology beyond the UK? E.g. studies by Hudson in the US

Epidemiology in the US has been added- thank you for the Hudson recommendation. “More recent research suggests that men could in fact represent as many as 1 in 5 people with EDs in the UK, with this number rising to 1 in 4 people with EDs in the US”

2. I believe there may have been a previous qualitative study of clinician views -?de Beer 2012 which could be cited (de Beer, Z., & Wren, B. (2012). Eating disorders in males. In J. Fox & K. Goss (Eds.), Wiley series in clinical psychology. Eating and its disorders (pp. 427-441). : Wiley-Blackwell.<http://dx.doi.org/10.1002/9781118328910.ch27>)

We have looked at the mentioned chapter and found that they included a qualitative study of men’s views, and included clinical recommendations rather than a full qualitative study on clinicians themselves? That previous research has consisted of recommendations rather than empirical research has been added, along with the de Beer citation: “Whilst previous literature has reflected on the need male treatment adaptations, these have not been empirically explored and rather represent clinical recommendations (de Beer, Z., & Wren, B. (2012).”

Method

1. Were there any constraints for the team leader who did the interviews – perhaps there could be a reflection on this and if the seniority imposed any constraints?

The following sentences have been added to reflect on this issue: “Whilst the impact of participants being interviewed by a senior staff member was a concern for the research team, steps were taken to minimise this as an issue: interviews were held in a room in the hospital separate from the department, and CN discussed the study with participants prior to the actual interview. Moreover, it

was emphasised that this was a service improvement project, with an emphasis on service provision rather than assessing the participants as individuals.”

2. Explain the role of The South London and Maudsley NHS trust governance committee. Was this study approved as part of a quality assurance project?

The nature of the approval has been made clearer: “The study was approved by South London and Maudsley NHS trust governance committee as part of a service improvement project assessing current treatment provision for men with EDs”.

Results

1. A minor point - was thematic saturation reached?

This point has been expanded- “Following the interviews with the 10 participants who had first agreed to participate, it was judged by the authors that data saturation had been reached as no new information was seen to be emerging from the interviews”.

Discussion

1. The statement “the finding of this study that these needs can be met by standard treatment approaches” is inaccurate as this was not a treatment study, rather “the finding of this study was that clinicians perceived that these needs can be met by standard treatment approaches”

Wording changed as recommended by reviewer

Reviewer: 2

Reviewer Name: Prof. Ulrich Voderholzer, MD

Institution and Country: Medical director, Schoen Clinic Roseneck, Prien am Chiemsee, Germany

Major comments

1. Introduction: In general, the introduction is well written, structured, and provides a reasonable framework for the present study. It is not fully clear, yet, how a qualitative study on clinician opinions could contribute to answer the questions that larger empirical studies left open. As a reader, the break from the last paragraph in the introduction to your aims is rather harsh. Wouldn't it be more intuitive to interview male ED patients if the aim is to explore specific treatment needs, as done by Robinson et al., 2003. Only a suggestion, but may be you could frame your aims in reference to this study.

The framing of the last paragraph has been altered to reflect that the aim of this study is to explore clinician views: “when men do access treatment in the UK, it is unclear how far current approaches used by clinicians require adaptation for this population”

2. Method: How were the 10 clinicians chosen? Randomly? Were there any male clinicians in the department (though, we later learn from participant 3 that there are none)? Please specify how the final selection came about.

The selection process has been elaborated: “Clinicians were invited to take part in an interview assessing their views on treating male EDs through email. All individuals agreed to participate, representing 70% of the clinicians in the department, and so formed the final cohort.”

That all clinicians in the department were female is reflected in the sentence “All participants were female, reflecting the all-female staff demography of clinicians within the department.”

3. Method, p 7: On what ground was it judged by the authors that data saturation had been reached?

This point has been expanded- “Following the interviews with the 10 participants who had first been invited to the study, it was judged by the authors that data saturation had been reached as no new information was seen to be emerging from the interviews”.

4. Results, general: I expected to find the four themes identified in the analysis as headers in the results section. As I understand, themes 2 and 3 pertaining to treatment differences and possible adaptations were combined in “treatment approaches”. While I think there is nothing wrong with this approach, it somewhat supersedes the structure proposed in the analysis section. I stumbled upon this inconsistency while reading.

The four themes mentioned by the reviewer refer to the categories used for coding the data, rather than the themes that emerged following analysis. That these are different has been made clearer: “This gave rise to a coding framework consisting of four main categories focusing on male/female symptom differences, male/female treatment differences, need for male treatment adaptations, and service improvements... Three themes emerged from the analysis following coding: male-specific issues, treatment approaches, and creating a male friendly environment.”

5. Discussion, p 12, ln 15: That first sentence in the discussion is an overstatement of your findings. What you could actually conclude based on your data is that (10 selected) clinicians suggest that men have specific treatment needs. Whether men actually have specific needs, can at best indirectly be examined with your study design. Please consider rephrasing. The following paragraph citing several quantitative studies, however, very well highlights how your findings add a new perspective to what is already known.

This has been qualified: “The findings of this study suggest that, from the perspective of the clinicians interviewed, men with EDs do have gender specific treatment needs.”

6. Discussion, p 13, lns 1-7: In this paragraph, I miss the notion that clinician openness and empathy are basic therapeutic skills that are instrumental in any psychotherapy for anybody with any disorder. Gender is only one of a range of characteristics that affect therapy. So would marital status, education, parenthood, language proficiency, cultural background etc. and all afford adaptation of the theoretical framework to the individual. Please elaborate on this aspect and be more specific what adaptations clinicians will need to make for MEN with EDs.

That this reflects basic therapeutic skills has been added, along with emphasising that subsequent recommendations are for men specifically: “This general emphasis on clinician openness and empathy reflects wider therapeutic principles that clinicians should be flexible in adapting therapy to individual needs, even whilst following manualised treatment programmes . However, the clinicians interviewed in this study highlighted a number of adaptations specifically for men with EDs (Addis, 1997).”

7. Discussion, p 14: The issue of male-only treatment setting is discussed in a balanced manner. It

is not fully clear to me, however, how your study could show that segregated treatment settings would be harmful. I actually agree with this notion and, besides that, it is just not possible to create male-only settings on a larger scale as most therapists are women. Also, I think that men could benefit from having a female therapist, whom they might expect to be more emotional. Hence, I think it would be more worthwhile to discuss here how existing facilities and treatment contents could be adapted to men. You alluded to the issue of body shape. Could you elaborate a bit more on how your study contributes to this aspect? Similarly to the very important paragraph on physical complications.

This section has been expanded on two points: “The current study indicates that gender-segregated treatment spaces, rather than helping patients, may in fact be harmful to the therapeutic process: clinicians felt that mixed-gender spaces, whether this meant a female therapist or mixed-gender groups, encouraged men to express emotion, and were concerned that men would feel inhibited in the company of only other men.”

“Therefore, this study emphasises the need to create a male-friendly treatment environment to combat this perception. For example, clinicians in this study highlighted the issue of making environmental aspects such as the waiting room more gender-neutral. Moreover, with quantitative research documenting that men with EDs may be more concerned with muscularity than an emphasis on thinness, treatment materials for EDs may require adaptation to include these diverse body image issues (Griffiths, Murray, & Touyz, 2015; Murray, Griffiths, & Mond, 2016; Murray et al., 2012). Moreover, these materials could be adapted to ensure that they are more gender neutral, such as by including examples and images of men.”

Minor comments

1. Abstract, p 2, ln 5: the sentence should read “incidence rate (...) in men is rising”.

Altered to “there is a growing body of evidence that the incidence rate of eating disorders in men is rising”

2. Abstract, p 2, ln 18: the sentence is rather long and could be partitioned, as it conveys the main message of the paper.

Partitioned as follows: “Male specific issues identified by participants included an increased focus on muscularity and difficulty expressing or discussing emotion. Clinicians also suggested that men may be more likely to adopt a performance based approach to recovery where patients focus on meeting goals or targets rather than addressing the issues underlying their illness.”

3. Introduction, p 3, lns 20-21: Style; repetition of “in fact”, which might not be necessary in any of the two sentences.

“In fact” deleted in both incidences.

4. Method, data collection, pp 6-7: the information that the interviews lasted 20 to 40 minutes is given twice

Repetition deleted.

5. Method, analysis, p 7, ln 15: “coded” instead of “coding” data?

Changed to coded

6. Method, patient involvement: I think it is a great feature of your study that patients had a saying in it. Could you may be give one or two examples of how these patients informed your study? Also, a paragraph should contain at least two sentences.

Paragraph expanded: “The development and design of this study was informed by a patient steering group consisting of men who had received treatment for EDs in the service. The notion that men may require treatment adaptations was initially explored with this group, with their responses leading to the development of the service improvement project.”

7. Results, p 8, ln 13: there are two commas missing, I think: “...not allowed to be, you know, weak and vulnerable...”

Punctuation altered as recommended

8. Discussion: not really a limitation but I think you should mention somewhere that you investigated EDs in general and not separately according to diagnosis. For example, you would expect women with AN to have similar difficulties expressing emotions that men with AN. May be you could speculate in one or two sentences about whether your results might apply more or less to the different EDs.

We have expanded the results to reflect clinician views on the individual disorders. “In particular, the emotional dysregulation symptoms associated with binge eating disorder (BED) were perceived to be similar across both men and women, enabling clinicians to approach treating individuals in a similar way:

“I think I’m doing a lot of the same work with binge eaters, yeah- I think a lot of it is focused on getting them to connect with how they feel. I think that once they’ve done that there’s a major turning point for both females and males. So I don’t think that there’s anything gender specific that I’m doing with them.” (Participant 2).

Consistent with this point, clinicians additionally highlighted that difficulty expressing emotion was not limited to male patients, and in fact was common across AN patients.”

In addition, we have added a sentence in limitations: “In addition, this study interviewed clinicians on their views on male EDs in general, rather than specific ED types: while the clinicians in this study did briefly discuss different ED types, including AN and BED, further research is needed to explore gender differences across EDs.”

VERSION 2 – REVIEW

REVIEWER	Phillipa Hay Western Sydney University, Australia
REVIEW RETURNED	19-Apr-2018
GENERAL COMMENTS	The authors have been very responsive -there are some minor typographical issues needing attention e.g. Murray et al needs a numbered reference in line 2 p 4.
REVIEWER	Ulrich Voderholzer Schoen Clinic Roseneck, Germany

REVIEW RETURNED	11-May-2018
GENERAL COMMENTS	The authors have carefully considered all concerns and fully addressed all questions. Personally I recommend the manuscript for publication now.